# A Phytomelatonin-Rich Extract Obtained from Selected Herbs with Application as Plant Growth Regulator

**DOI:** 10.3390/plants10102143

**Published:** 2021-10-09

**Authors:** Josefa Hernández-Ruiz, Antonio Cano, Marino B. Arnao

**Affiliations:** Department of Plant Biology (Plant Physiology), Faculty of Biology, University of Murcia, 30100 Murcia, Spain; jhruiz@um.es (J.H.-R.); aclario@um.es (A.C.)

**Keywords:** biostimulator, green chemistry, melatonin, phytomelatonin, plant protector, plant stress

## Abstract

The animal hormone melatonin (N-acetyl-5-methoxytryptamine) is a pleiotropic molecule with multiple and various functions. Phytomelatonin is the melatonin from plants and was discovered in 1995 in some species. Phytomelatonin is considered an interesting molecule in the physiology of plants, as it seems to be involved in many actions, such as germination, growth, rooting and parthenocarpy, including fruit set and ripening; it also seems to play a role during postharvest. It has been studied in processes such as primary and secondary metabolism, photosynthesis and senescence, as well as in the nitrogen and sulfur cycles. Phytomelatonin up- and down-regulates many relevant genes related to plant hormones and key genes related to the above-mentioned aspects. One of the most decisive aspects of phytomelatonin is its relevant role as a bioprotective and alleviating agent against both biotic and abiotic stressors, which has opened up the possibility of using melatonin as a phytoprotector and biostimulant in agriculture. In this respect, using material of plant origin to obtain extracts rich in phytomelatonin instead of using synthetic melatonin (thus avoiding unwanted by-products) has become a topic of discussion. This work characterized the phytomelatonin-rich extracts obtained from selected herbs and determined their contents of phytomelatonin, phenols and flavonoids; the antioxidant activity was also measured. Finally, two melatonin-specific bioassays in plants were applied to demonstrate the excellent biological properties of the natural phytomelatonin-rich extracts obtained. The herb composition and the protocols for obtaining the extracts rich in phytomelatonin are in the process of registration for their legal protection.

## 1. Introduction

Phytomelatonin (N-acetyl-5-methoxytryptamine) was discovered in plants in 1995 [1,2,3] and has since been detected in practically all plant species analyzed, although the actual content varies widely with the plant species [4]. In general, aromatic medicinal plants seem to have high phytomelatonin levels [5]. Relevant factors in this respect, which affect the endogenous phytomelatonin content, are the growth and development conditions. Many physiological studies have been carried out on the role of phytomelatonin in plants, observing that it acts as a protective biomolecule capable of counteracting physical, chemical and biological stressors. Indeed, all the common stress agents studied in plants, whether of a physical (heat, cold and ultraviolet radiation), chemical (salinity, alkalinity, drought, waterlogging, heavy metals, mineral deficit/excess, pesticides and combinations of the same) or biotic (virus, bacteria and fungi) nature, induce an increase in the endogenous phytomelatonin level, activating the respective anti-stress response and increasing the tolerance to the stressor [6,7,8,9,10,11,12,13]. In addition to its protective actions against stressors, other interesting physiological functions have been attributed to phytomelatonin, including processes such as germination, rooting, seedling growth, flowering, parthenocarpy, fruit set and ripening. Other aspects related to phytomelatonin that have been studied include its role in photosynthesis, ripening and senescence, the primary metabolism (carbohydrates, lipid, amino acids, nitrogen, sulfur) and secondary metabolism (flavonoids, anthocyanins, carotenoids, essential oils), including osmoregulation, and the regulation of the plant hormones auxin (IAA), gibberellins (GAs), cytokinins, abscisic acid (ABA), ethylene (ET), jasmonates (JA), salicylic acid (SA), polyamines, brassinosteroids and strigolactones [14,15,16,17,18,19,20,21,22,23,24,25,26,27,28,29].

The phytomelatonin biosynthesis pathway is well characterized, and was shown to be different from animal cells in some steps [30,31], according to the following scheme (Figure 1) [16,32].

In plants, phytomelatonin induces many changes in gene expression, resulting in a global biostimulatory action [14]. Due to the diversity of its actions, phytomelatonin was proposed not only as a plant master regulator, but also as a novel plant hormone, since its receptor (PMTR1), a plasma membrane receptor that interacts with G-proteins, was identified in Arabidopsis [33]. Phytomelatonin acts as a regulator of the redox network in plants, both directly and indirectly regulating the oxygen- and nitrogen-radical species (ROS-RNS) levels and the expression of many regulation factors through the nitric oxide and hydrogen peroxide signaling cascades, among other pathways. Hence, phytomelatonin can regulate redox homeostasis, balancing ROS-RNS and the expression of related key enzymes (NOS-like, NR, RbOHs, ASA-GSH cycle, antioxidant enzymes such as catalases, peroxidases, superoxide dismutases, etc.) [15].

Its role as a stimulator of systemic acquired resistance (SAR) is noteworthy, favoring the health of crops in possible bacterial, fungal and viral pathogenic infections. Phytomelatonin has been used as an alleviating agent in several fungal diseases caused by *Botrytis cinerea, Phytophthora infestans, Phytophthora nicotianae, Plasmodiophora brassicae, Penicillium digitatum, Fusarium* spp. and *Alternaria* spp. [11,18,34]. Recently, it was demonstrated that exogenous melatonin treatment is able to activate the plant defenses known as ETI (effector-triggered immunity) and PTI (PAMP: pathogen-associated molecular pattern-triggered immunity) in watermelon and *Arabidopsis*, according to transcriptomic data [35]. In the face of phytobacterial infections, phytomelatonin increases the resistance in the rice bacterial leaf streak caused by *Xanthomonas oryzae* pv. *oryzicola* (Xoo) [36,37]. Exogenous melatonin treatment revert the disorder *Candidatus Liberibacter asiaticus* in citrus (greening disease) [38]. Additionally, melatonin treatments confer plant disease resistance against bacterial blight in cassava [39,40,41]. In general, melatonin induces plant resistance through the upregulation of defense genes such as plant defensin (PDF) and plant resistance (PR), while also upregulating SA-, ET- and JA-associated genes and mitogen-activated protein kinase (MAPK) cascades [11,16]. In the case of viruses, exogenous melatonin results in the shortening of tobacco mosaic virus (TMV), viral RNA and virus particles in infected *Nicotiana glutinosa* and *Solanum lycopersicum* plants [42]. Moreover, melatonin efficiently decreases apple stem grooving virus (ASGV) from the in vitro virus-infected apple shoots and so is used to generate virus-free plants [43]. In addition, in rice, melatonin treatments increase resistance against RSV (rice stripe virus), which is one of the most pathogenic rice viruses in East Asia [44].

Melatonin has been widely studied and its protective and biostimulator role in plants has been clearly established. However, the use of synthetic melatonin is common in lab experimentation. The possible use of plant material as a source of phytomelatonin-rich extracts instead of synthetic melatonin (thus avoiding unwanted by-products) has been discussed [45,46,47]. It should be remembered that melatonin is a hormone in animals and humans, but it is also present in all other living beings, including prokaryotes, eukaryotes, algae, fungi, insects and other animals, both aquatic and terrestrial. The possible use of synthetic melatonin in agriculture was widely discussed recently in an extensive review, taking into account the legislation of the European Community (EC). In that review, the use of natural extracts rich in phytomelatonin, with clear advantages and some disadvantages, was proposed as an alternative [48].

In this work, several phytomelatonin-rich extracts obtained from selected herbs were characterized, and the main biochemical components they contain were determined. The different phytomelatonin contents of several types of herb extracts were measured by LC-fluorescence and LC-QTOF/MS. Their richness in phenols, flavonoids and antioxidant activity was analyzed. Additionally, two melatonin bioassays carried out in plants confirm the excellent biological properties of the natural phytomelatonin-rich extracts obtained. The objective of this article is to propose the use of herb extracts rich in phytomelatonin as biostimulants and plant protectors in crops and in biotechnology.

## 2. Materials and Methods

### 2.1. Chemicals

The chemicals, solvents (methanol, ethanol and ethyl acetate) and reagents used were from Sigma-Aldrich (Madrid, Spain). Milli-Q system (Milli-Q Corp, Merck KGaA, Darmstadt, Alemania) ultra-pure water was used.

### 2.2. Plant Material

A selection of herbs was used. The selected herbs (SHB) and the extractive protocols are in the process of registration for their legal protection, so their composition cannot be revealed. SHB were treated according to the scheme in Figure 2. From SHB, three different concentrated final products, so-called Phytomel-Agro (PTMA), were obtained. Both SHB and PTMAs were analyzed to measure different chemical and biochemical parameters.

### 2.3. Proximate Analysis

Samples of SHB and PTMAs were analyzed for moisture (Method no. 945.15), ash (Method no. 942.05), crude protein (Kjeldahl method using a factor of 6.25, Method no. 920.54), crude fat (Method no. 920.39) and dietary fiber (Method no. 985.29) contents, according to the AOAC methods [49]. The data are expressed as % of fresh material.

### 2.4. Biochemical Analysis

Total phenolic and flavonoid content, and hydrophilic antioxidant activity in SHB and PTMAs were determined. Folin–Ciocalteu’s reagent was applied to determine the total content of phenolic compounds [50]. Five hundred microliters of the sample was placed in a glass test tube, and 0.85 mL of water, 50 µL of 1N NaOH and 50 µL of Folin–Ciocalteu’s reagent were added. The reaction medium was allowed to react in the dark, at 30 °C for 1 h, before the absorbance at 755 nm was measured with a UV-Vis spectrophotometer (Perkin-Elmer, model Lambda 2S, Hamburg Germany). The results are expressed as moles of gallic acid (used as standard) equivalents per gram of dry matter (DW) or oleoresin (ORS).

To measure the total flavonoid content, the aluminum chloride colorimetric method was applied [50,51]. Quercetin was used as a standard. Quercetin (10 mg) was dissolved in 80% ethanol and then diluted to 12.5, 25, 50, 75 and 100 µg/mL. The diluted standard solutions were separately mixed with 50 µL of 10% aluminum chloride, 50 µL of 1 M potassium acetate and 0.85 mL of distilled water. After incubation at 30 °C for 30 min, the absorbance of the reaction mixture was measured at 415 nm. A similar procedure was applied to SHB and PTMAs samples for total flavonoid content analysis. The results are expressed as moles of quercetin equivalents per gram of dry matter (DW) or oleoresin (ORS).

Hydrophilic antioxidant activity was measured in the samples using the method described in [52,53], which is based on the ability of the antioxidants of a sample to reduce the radical cation of 2,2′-azino-bis-3-(ethylbenzothiazoline-6-sulphonic acid) (ABTS·+), determined by the discoloration of ABTS·+ and measuring the quenching of the absorbance at 730 nm. This activity was calculated by comparing the values of the sample with a standard curve of ascorbic acid and is expressed as moles of ascorbic acid equivalents per gram of dry matter (DW) or oleoresin (ORS).

### 2.5. Phytomelatonin Analysis

Phytomelatonin content in SHB and PTMAs was determined by liquid chromatography (LC) with fluorescence detection and by LC with time-of-flight mass spectroscopy (LC-QTOF/MS) as in [7,54,55]. A Jasco liquid chromatograph Serie-2000 (Tokyo, Japan) equipped with an online degasser, binary pump, auto sampler, thermo-stated column and a Jasco FP-2020-Plus fluorescence detector were used to measure phytomelatonin levels. An excitation wavelength of 280 nm and an emission wavelength of 350 nm were selected. A Waters Spherisorb-S5 ODS2 column (SigmaAldrich, Spain)(250 × 4.6 mm) was used. The isocratic mobile phase consisted of water:acetonitrile (80:20) at a flow rate of 0.2 mL/min. The data were analyzed using the Jasco ChromNAV v.1.09.03 Data System Software (Tokyo, Japan). Forcorrect identification, an in-line fluorescence spectral analysis (using the Jasco Spectra Manager Software (Tokyo, Japan) compared the excitation and emission spectra of standard melatonin with the corresponding peak of phytomelatonin in the samples [54].

Identification of phytomelatonin in plant extracts was also confirmed using an LC/MS system consisting of an Agilent 1290 Infinity II Series LC (Agilent Technologies, Santa Clara, CA, USA) equipped with an Automated Multisampler module and a High-Speed Binary Pump, and connected to an Agilent 6550 Q-TOF Mass Spectrometer (Agilent Technologies, Santa Clara, CA, USA) using an Agilent Jet Stream Dual electrospray (AJS-Dual ESI) interface (Santa Clara, CA, USA). Experimental parameters for HPLC and Q-TOF were set in MassHunter Workstation Data Acquisition software (Agilent Technologies, Rev. B.08.00, Santa Clara, CA, USA) [55].

Samples (SHB and PTMAs) were filtered through 0.2 µm filters and Sep-Pack C18 filtered before analyzing. Standards or samples (20 µL) were injected onto a Waters XBridge C18 5 µm, 100 2.1 mm LC column (SigmaAldrich, Spain), at a flow rate of 0.4 mL/min, and thermo-stated at 40 °C. Solvents A (MilliQ water with 0.1% formic acid) and B (acetonitrile with 0.1% formic acid) were used for the compound separation with an initial condition of 95% solvent A and 5% solvent B. After injection, the initial conditions were maintained for 2 min, and then, compounds were eluted using a linear gradient 5–100% solvent B for 8 min. A hundred percent solvent B was maintained for 2 min, and the system was finally equilibrated at 5% solvent B for 3 min before a new injection.

The mass spectrometer was operated in the positive mode. The nebulizer gas pressure was set to 40 psi, and the drying gas flow was set to 13 L/min at a temperature of 250 °C. The sheath gas flow was set to 12 L/min at a temperature of 300 °C. The capillary spray, nozzle, fragmentor and octopole 1 RF Vpp voltages were 3500 V, 50 V, 150 V and 750 V, respectively. Profile data in the 50–300 *m*/*z* range were acquired for MS scans in 2 GHz extended dynamic range mode. A reference mass of 121.0509 was used. Data analysis was performed with MassHunter Qualitative Analysis Navigator software (Agilent Technologies, Rev. B.08.00, Santa Clara, USA. The signal corresponding to phytomelatonin was extracted and quantified with an *m*/*z* of 233.1285.

### 2.6. Biological Assay by Dark-Induced Senescence of Leaves

The bioassay was developed by Arnao et al. [6] and based on the protective effect of melatonin against chlorophyll degradation during the dark-induced senescence of leaves. Briefly, sterilized parsley (*Petroselinum crispum*) leaves (0.3 g FW) were incubated in 10 mM potassium phosphate buffer (pH 6.0) in the presence of synthetic melatonin (0.1 and 1 mM) and PTMA at different concentrations. Eight leaves were used in each treatment, which was repeated 4 times. The chlorophylls retained in the leaves in each treatment were analyzed after 5 days at 24 °C in darkness following the method described by Lichtenhaler and Wellburn [56].

### 2.7. Biological Assay by Cotyledon Growth in Darkness

The bioassay based on the melatonin-induced growth of lupin (*Lupinus albus*) cotyledons was developed by Hernández-Ruiz et al. [57]. Briefly, fully imbibed etiolated lupin cotyledons, without the embryo, were incubated with synthetic melatonin and PTMAs. The area and fresh weight of the cotyledons in each treatment were recorded initially and after 48 h of incubation at 24 °C in darkness. Eight cotyledons were used in each treatment, which was repeated 4 times. The increase in the area of lupin cotyledons was measured using the software application Adobe Photoshop (San José, CA, USA).

### 2.8. Statistical Analysis

Statistical approaches were applied using the SPSS 10 program (SPSS Inc., Chicago, IL, USA), using the LSD multiple range test to establish significant differences at *p* < 0.05. The results are expressed as mean ± standard error (SE, n = 4).

## 3. Results and Discussion

A mixture of selected herbs (SHB) was used to obtain phytomelatonin-rich extracts (PTMAs) (Figure 2). The protocols are currently in the process of registration for their legal protection. These plants were from non-transgenic seeds and obtained by organic culture. The green-extractive process applied permits to obtain stabilized and natural phytomelatonin-rich extracts ready for use. The selected herbs (SHB) in a dried form were subjected to a solid–liquid extraction step. Next, two different protocols were applied (type I and II). In type I, only a simple filtration and evaporation of the solvent was applied (Phytomel-Agro1). In type II, in addition to the filtration and evaporation steps, two modifications were introduced: a centrifugation step for Phytomel-Agro2and an ultrasound treatment step for Phytomel-Agro3.

Table 1 shows the results of the proximate analysis of the plant material (SHB) and the final extracts (PTMAs). A high protein, carbohydrate and fiber content can be observed in SHB; by contrast, low oil content is presented. An intensive concentration process to obtain high phytomelatonin content in PTMAs was applied (Figure 2).

Logically, the concentration process quantitatively affected all the parameters measured (Table 1). Thus, plant oils (crude fat) were the main component of PTMA2 and PTMA3, in which phytomelatonin is perfectly solubilized and extracted.

Figure 3 shows a representative chromatograms (by fluorescence detection and by Q-TOF/MS) of a concentrated extract (PTMAs), according to our previously published data [54,55].

Table 2 shows the phytomelatonin contents of the initial plant material (SHB) and of the final concentrated product (PTMAs). Using these protocols, significant amounts of phytomelatonin can be extracted and concentrated from these plants; between ~35–50 µg of phytomelatonin by gram of oleoresin (ORS) in PTMA-2 and PTMA-3 can be obtained. Regarding the ratio between the plant material necessary to obtain a quantity of PTMA3, about 200–250 g DW of plants is needed to obtain 50 µg phytomelatonin. This amount of phytomelatonin can be considered sufficient to prepare solutions with which to treat agronomic plants [43].

Table 3 shows the values of other biochemical parameters, such as total phenolic and flavonoid contents and hydrophilic antioxidant activity. As is to be expected, the SHB showed lower values of antioxidant activity and total phenols and flavonoids than PTMAs due to the concentrating process applied. One of the most studied properties of melatonin is its quality as a natural antioxidant. Although its effects as a biostimulator go much further than that of a mere antioxidant [15], the fact that phytomelatonin is found in a matrix with a considerable number of antioxidant molecules, such as phenols and flavonoids, ensures a high antioxidant activity (AAH) (see Table 3), which preserves it. Biochemically, phenols and flavonoids possess multiple beneficial properties for plants, which add to the natural quality of the phytomelatonin-rich extracts (PTMAs) [58].

To find out if our obtained phytomelatonin-rich extracts had functional properties, we applied them to two previously developed melatonin bioassays. In the first bioassay, the senescence of parsley leaves treated with the melatonin solutions slowed down, as estimated from the chlorophyll lost in leaves after dark incubation. Figure 4 shows this effect of both PTMA3 extract and synthetic melatonin (sMEL) at different concentrations. As can be seen, the response was concentration dependent, with the optimal response obtained at 1 mM sMEL, similarly to that observed in the original bioassay in barley leaves [6]. As can be seen, the PTMA3 extract also produced a similar response assMEL, which means that our extracts were physiologically active in the bioassay.

The bioassay of the melatonin-induced growth of lupin cotyledons was based on the growth-induction capacity of melatonin in plant tissues such as roots, cotyledons, epicotyls and hypocotyls [57,59,60,61]. Figure 5 shows the effect of synthetic melatonin and PTMA3 on cotyledon area expansion, as well as the increases in fresh weight, after 48 h of incubation at 24 °C in darkness. As can be seen, the cotyledon area and fresh weight increased with respect to the control incubation (buffer medium only) for both treatments. Moreover, the phytomelatonin-rich extract PTMA3 showed an excellent physiological behavior, stimulating cotyledon growth in a similar way to synthetic melatonin (sMEL), according to our pioneering bioassay [57]. Of the three extracts obtained, PTMA3 is the richest in phytomelatonin and, therefore, more active in the tests applied in Figure 4 and Figure 5. PTMA1 did not present a response in the applied tests, probably due to its low level of phytomelatonin, or because it is a vast and more complex extract. Regarding PTMA2, we observed activity with less intensity in the tests, and therefore we focused on the most active extract, PTMA3.

According to our previous studies on the potential of melatonin on the inhibition of chlorophyll degradation (test in Figure 4) and on cotyledon growth (test in Figure 5), auxin and cytokinins are plant hormones that can test positive. Similarly, it seems that ABA affects it in a negative way [6]. For this reason, we have never maintained that phytomelatonin can, exclusively, be the molecule responsible for these effects in physiological conditions. In this sense, the role of phytomelatonin on foliar senescence was demonstrated. Melatonin down-regulates several chlorophyll degrading genes through the senescence factor SAG12, among others. The melatonin action is possibly cytokinin-mediated because melatonin up-regulates cytokinin level and some signaling elements inhibit the senescent process in a co-active melatonin–cytokinin collaboration [24]. In any case, we found a natural extract that, being rich in phytomelatonin, can behave with excellent regulatory properties of plant development, which can be applied to crops. However, the complete hormonal characterization of PTMA3 is one of our immediate goals.

Phytomelatonin contents in wild plants and also in edible plants range from nanograms to micrograms per gram of plant tissue. Generally, seeds, leaves, stems, seedlings and roots present the highest phytomelatonin levels, and fruits the lowest. Medicinal aromatic plants have significantly higher levels than seeds and fruits. Additionally, growing conditions and harvesting, among other factors, affect phytomelatonin level in plant tissues [4,32]. After an arduous search and analysis of a multitude of aromatic medicinal plants, we selected a group of herbs (SHB) to obtain PTMAs, which are rich in phytomelatonin. Our process at the lab scale is laborious but provides concentrated extracts that are very rich in phytomelatonin and available for use as an active material in natural agro-phytochemical treatments as a plant growth regulator. In any case, our next challenge is to obtain new extracts that are even richer in phytomelatonin and to make it economically profitable on a large scale.

## 4. Concluding Remarks

In conclusion, the SHB and green-extractive processes described in this work enabled us to obtain natural phytomelatonin-rich extracts (PTMAs), free of unwanted by-products. In this case, the optimal phytomelatonin content of PTMAs was around 50 µg of phytomelatonin by gram of oleoresin, reaching a sufficient concentration to be used as an initial material in a variety of agricultural and biotechnological applications. Generally, crop treatments with plant growth regulators or biostimulators may involve concentrations in the order of 5–100 µM. The PTMAs obtained in this study are concentrated enough to avoid any problems related to obtaining solutions for the massive treatment of plants. Furthermore, the amphiphilic nature of phytomelatonin molecules means that they are easily absorbed either by the roots or by the leaves. However, our group is engaged in conducting stability, absorption and transport tests of the active substance in various crops. The possible use of synthetic melatonin as a chemical in crops requires more exhaustive studies for its possible legalization in different countries. However, the use of natural extracts rich in phytomelatonin can be considered as an alternative with possibly fewer legal obstacles.

## Figures and Tables

**Figure 1 plants-10-02143-f001:**
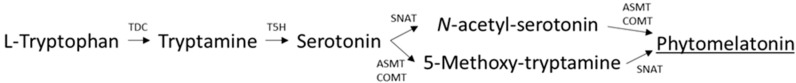
Simple scheme of phytomelatonin biosynthesis pathway in plants. The steps are catalyzed by the enzymes TDC, T5H, SNAT and COMT.

**Figure 2 plants-10-02143-f002:**
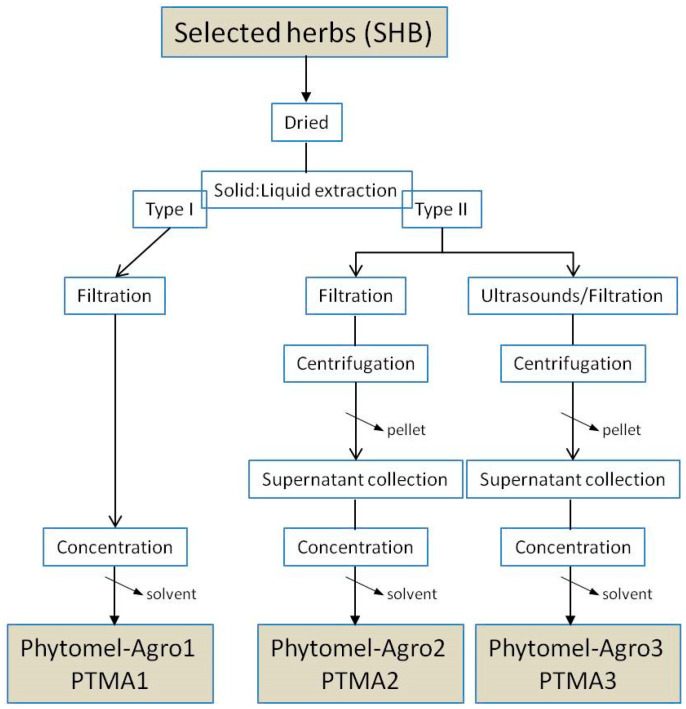
Scheme of the protocol used to obtain phytomelatonin-rich extracts (PTMAs) from selected herbs (SHB).

**Figure 3 plants-10-02143-f003:**
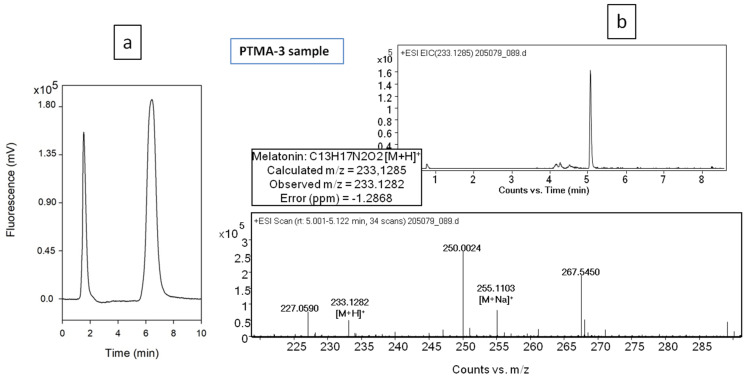
(**a**) shows the representative chromatogram of a PTMA sample with liquid chromatography and fluorescence detection. Fluorescence detection was programmed at λexc of 280 nm and λemi of 350 nm. The melatonin peak has a retention time of 6.5 min. (**b**) shows the representative chromatogram and mass spectra of PTMA3 sample using liquid chromatography with time-of-flight/mass spectrometry (LC-QTOF/MS).

**Figure 4 plants-10-02143-f004:**
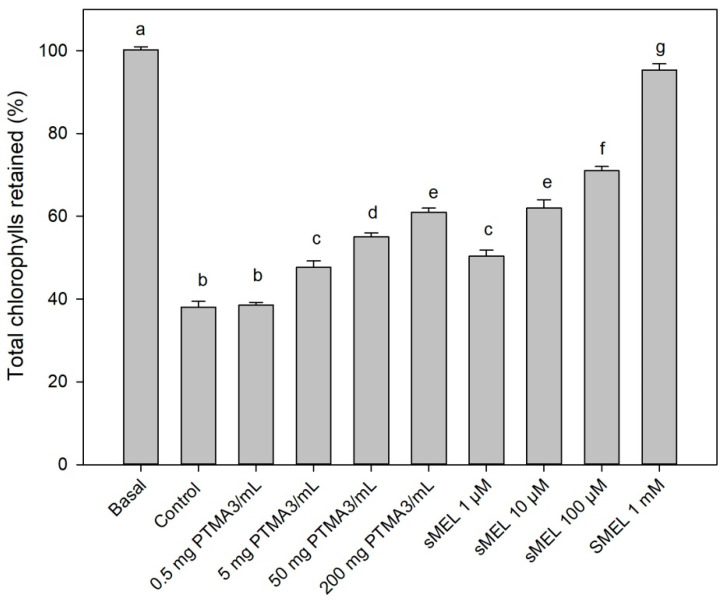
Effect of PTMA3 and synthetic melatonin (sMEL) on the total chlorophyll retained in parsley leaves incubated in a buffered medium and a control medium (only buffer) in darkness. Basal (day 0) corresponds to fresh leaves not treated. Error bars represent standard errors of the mean of four replicate experiments (n = 4). Different superscript letters indicate statistically significant differences at *p* < 0.05.

**Figure 5 plants-10-02143-f005:**
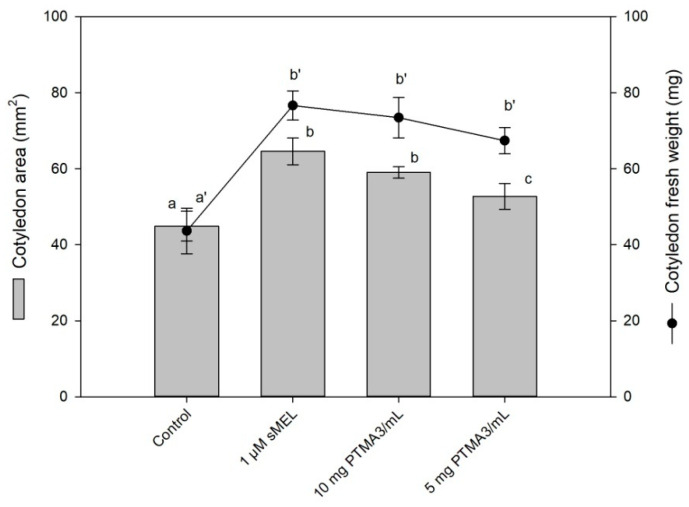
Effect of synthetic melatonin (sMEL) and PTMA3 on etiolated lupin cotyledon area and fresh weight after 48 h of incubation. Control (only buffer media). Data represent the mean values for each treatment. Error bars represent standard errors of the mean of four replicate experiments (n = 4). Different superscript letters indicate statistically significant differences at *p* < 0.05.

**Table 1 plants-10-02143-t001:** Proximate analysis of selected herbs (SHB) and the final products (PTMAs) (±standard errors of the mean of four replicate experiments).

Components (in %, as Fresh Material)	SHB	PTMA1	PTMA2	PTMA3
Moisture	89.61 ± 4.36	25.50 ± 2.28	11.61 ± 0.89	10.22 ± 0.81
Ash	0.37 ± 0.02	2.18 ± 0.12	Trace	Trace
Crude proteins	3.50 ± 0.12	28.12 ± 2.11	5.72 ± 0.20	6.12 ± 0.40
Crude fat	0.55 ± 0.04	10.23 ± 1.09	54.84 ± 3.14	57.36 ± 3.82
Dietary fiber	2.34 ± 0.08	16.92 ± 1.58	0.97 ± 0.06	1.15 ± 0.07
NFEM* (~carbohydrates)	3.63 ± 0.22	17.05 ± 1.15	26.86 ± 0.41	25.15 ± 0.37

NFEM*, nitrogen-free extractive material.

**Table 2 plants-10-02143-t002:** Phytomelatonin analysis of selected herbs (SHB) and the final concentrated products (PTMAs) (±standard errors of the mean of four replicate experiments).

Material	Phytomelatonin Content
SHB	0.35 ± 0.02 µg/g DW
PTMA1	11.7 ± 0.9 µg/g ORS
PTMA2	35.0 ± 2.9 µg/g ORS
PTMA3	50.2 ± 3.7 µg/g ORS

DW, dry weight; ORS, oleoresin.

**Table 3 plants-10-02143-t003:** Specific analysis of selected herbs (SHB) and the final products (PTMAs).

Parameter	SHB	PTMA1	PTMA2	PTMA3
Total phenolic content (TPC) (eq. gallic acid/g)	243.1 ± 21.2 nmoles/g DW	13.6 ± 1.1 µmoles/g ORS	261.5 ± 21.7 µmoles/g ORS	288.1 ± 23.0 µmoles/g ORS
Total flavonoid content (TFC) (eq. quercetin/g)	56.5 ± 3.9 nmoles/g DW	4.4 ± 0.3 µmoles/g ORS	90.4 ± 8.1 µmoles/g ORS	95.3 ± 7.4 µmoles/g ORS
Hydrophilic antioxidant activity (HAA) (eq. ascorbic acid/g)	172.7 ± 9.6 nmoles/g DW	9.3 ± 0.9 µmoles/g ORS	152.6 ± 12.5 µmoles/g ORS	162.7 ± 13.7 µmoles/g ORS

DW, dry weight; ORS, oleoresin.

## Data Availability

Not applicable.

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
