# Peer review of "A Phytomelatonin-Rich Extract Obtained from Selected Herbs with Application as Plant Growth Regulator"

_plants, 2021, doi:10.3390/plants10102143_

Round 1

Reviewer 1 Report

The work entitled "A phytomelatonin-rich extract obtained from selected

 herbs with application as plant growth regulator " is a paper whose information is novel, methodologically correct and whose results and conclusions are acceptable. From this point of view it is a paper that could be accepted for publication, but I have the following doubt:

The aim of this work and the conclusion it draws is to use plant extracts as a source of natural phytomelatonin instead of synthetic melatonin. Given that the selected herbs (SHB) and the extractive protocols are being  registered for their legal protection, the initial quantities and the oleoresin extracts obtained are not known, so it is not possible to judge whether it is feasible and sustainable. It may be that an acceptable amount of phytomelatonin can be obtained in the extracts, but to achieve this would require a very large initial quantity of plant material, and this would not be feasible. I think the editor should consider whether to publish this article without this information. Although the protocol is not specified, I believe that the ratio of plant material needed to obtain the indicated amount of phytomelatonin should be added.

Author Response

Rev1

Dear Reviewer, thank for your comments.

The work entitled "A phytomelatonin-rich extract obtained from selected herbs with application as plant growth regulator " is a paper whose information is novel, methodologically correct and whose results and conclusions are acceptable. From this point of view, it is a paper that could be accepted for publication, but I have the following doubt:

The aim of this work and the conclusion it draws is to use plant extracts as a source of natural phytomelatonin instead of synthetic melatonin. Given that the selected herbs (SHB) and the extractive protocols are being registered for their legal protection, the initial quantities and the oleoresin extracts obtained are not known, so it is not possible to judge whether it is feasible and sustainable. It may be that an acceptable amount of phytomelatonin can be obtained in the extracts, but to achieve this would require a very large initial quantity of plant material, and this would not be feasible. I think the editor should consider whether to publish this article without this information. Although the protocol is not specified, I believe that the ratio of plant material needed to obtain the indicated amount of phytomelatonin should be added.

Answer:

We understand the existing gap regarding the data on the plant material. We appreciate the understanding of the Rev1 since when dealing with a subject susceptible of exploitation we must safeguard the intellectual rights before their legal protection. However, as Rev1 suggested, some data can be revealed. The solvents used are conventional, including water, methanol, ethanol, and ethyl acetate. Regarding the ratio between the plant material necessary to obtain the quantity in PTMA3, it is about 200-250 g DW of plants to obtain 50 µg phytomelatonin. These data have been incorporated in the new version, lines 163 and 289-291.

Reviewer 2 Report

Dear Researchers

General remarks

The manuscript entitled: „A phytomelatonin-rich extract obtained from selected herbs with application as plant growth regulator by Josef Hernández-Ruiz, Antonio Cano, Marino B. Arnao concerns the analysis of chemical, biochemical and biological properties of raw plant material (the selected herbs (SHB)) and 3 types of extracts made of this material (PTMA1, PTMA2, PTMA3). The work contains interesting data that has been presented in a clear and legible way. The cognitive significance of this work consists in developing a method of extracting phytochemicals from raw plant material in terms of high phytomelatonin content. In addition, these studies are of great application importance as they constitute a preliminary stage for the production of this type of natural biostimulators and plant protectors on a larger scale. In my opinion, this work has great substantive and scientific value, but the chapter Materials and Methods should be supplemented.

The main drawback of this work is the lack of characteristics of the raw plant material that was used to make the extracts (species, variety, organ, origin of these plants, growing conditions). The method of preparing extracts from this material (type of solvent) should also be specified. In scientific publications, this type of information is necessary because they enable the repetition of these experiments and in the same or a different arrangement.

Detailed remarks

Lines 231-232 „  between  ~80-100  μ g  of  phytomelatonin  by  gram  of 231

oleoresin  (ORS)  in  PTMA-2  and  PTMA-3  can  be  obtained”  From the data in Table 2 it can be seen that these amounts are lower, i.e. 35.0 for PTMA2 and 50.2 for PTMA3.

Line 242 „The presence of all these natural compounds stabilizes final extracts, also  contributing to preserve at phytomelatonin molecules” As this statement does not follow from the presented data, please cite the relevant literature.

Why the authors did not present the results for the remaining extracts (PTMA1, PTMA3) in Figures 4 and 5

Line 290 „optimal phytomelatonin content of PTMAs was around 100 μg of phytomelatonin by gram of oleoresin” - the data presented in the table shows that these values are: approx. 50 μg of phytomelatonin by gram of oleoresin!?

Does the observed biological effect (inhibition of chlorophyll degradation during the dark-induced senescence of parsley leaves and stimulation of growth of lupin cotyledons) result from the presence of other plant hormones in the extracts?

Author Response

Rev 2

Dear Reviewer, thank for your comments.

General remarks

The manuscript entitled: „A phytomelatonin-rich extract obtained from selected herbs with application as plant growth regulator by Josef Hernández-Ruiz, Antonio Cano, Marino B. Arnao concerns the analysis of chemical, biochemical and biological properties of raw plant material (the selected herbs (SHB)) and 3 types of extracts made of this material (PTMA1, PTMA2, PTMA3). The work contains interesting data that has been presented in a clear and legible way. The cognitive significance of this work consists in developing a method of extracting phytochemicals from raw plant material in terms of high phytomelatonin content. In addition, these studies are of great application importance as they constitute a preliminary stage for the production of this type of natural biostimulators and plant protectors on a larger scale. In my opinion, this work has great substantive and scientific value, but the chapter Materials and Methods should be supplemented.

 The main drawback of this work is the lack of characteristics of the raw plant material that was used to make the extracts (species, variety, organ, origin of these plants, growing conditions). The method of preparing extracts from this material (type of solvent) should also be specified. In scientific publications, this type of information is necessary because they enable the repetition of these experiments and in the same or a different arrangement.

Answer:

We understand the existing gap regarding the data on the plant material. We appreciate the understanding of the Rev1 since when dealing with a subject susceptible of exploitation we must safeguard the intellectual rights before their legal protection. However, as Rev1 suggested, some data can be revealed. The solvents used are conventional, including water, methanol, ethanol, and ethyl acetate. Regarding the ratio between the plant material necessary to obtain the quantity in PTMA3, it is about 200-250 g DW of plants to obtain 50 µg phytomelatonin. These data have been incorporated in the new version, lines 163 and 289-291.

Detailed remarks

Lines 231-232 (in R1:L288), “between ~80-100 μg of phytomelatonin by gram oleoresin (ORS) in PTMA-2 and PTMA-3 can be obtained”. From the data in Table 2 it can be seen that these amounts are lower, i.e. 35.0 for PTMA2 and 50.2 for PTMA3.

Answer:

Unfortunately, an error in the text has occurred. Indeed, the correct data are those of table 2. The error has been corrected in the new version.

Line 242 (R1: L301)„The presence of all these natural compounds stabilizes final extracts, also contributing to preserve at phytomelatonin molecules” As this statement does not follow from the presented data, please cite the relevant literature.

Answer:

One of the most studied properties of melatonin is its qualities as a natural antioxidant. Although its effects as a biostimulator go much further than that of a mere antioxidant (ref. 58), the fact that phytomelatonin is found in a matrix with a considerable number of antioxidant molecules such as phenols and flavonoids, ensures a high antioxidant activity (AAH) (see table 3), which preserves it. This explanation and a new reference on the antioxidant qualities of phenols, flavonoids have been incorporated in the new version (lines 301-307).

Why the authors did not present the results for the remaining extracts (PTMA1, PTMA3) in Figures 4 and 5.

Answer:

Of the three extracts obtained, PTMA3 is the richest in phytomelatonin and, therefore, more active in the tests applied in Figs. 4 and 5. PTMA1 does not present a response in the applied tests, probably due to its low level of phytomelatonin or because it is a vast and more complex extract. Regarding PTMA2, we have observed activity with less intensity in the tests, and therefore we have focused on the most active extract, PTMA3. In any case, our next challenge is to obtain new extracts that are even richer in phytomelatonin. These considerations have been added to the text (Lines 339-344 and 370-371).

Line 290 (R1:L375) „optimal phytomelatonin content of PTMAs was around 100 μg of phytomelatonin by gram of oleoresin” - the data presented in the table shows that these values are: approx. 50 μg of phytomelatonin by gram of oleoresin!?

Answer:

As in line 288, the error has been corrected in the new version.

Does the observed biological effect (inhibition of chlorophyll degradation during the dark-induced senescence of parsley leaves and stimulation of growth of lupin cotyledons) result from the presence of other plant hormones in the extracts?

Answer:

According to our previous studies on the potential of melatonin on the inhibition of chlorophyll degradation (test in Fig 4) and on cotyledon growth (test in Fig 5), auxin and cytokinins are plant hormones that can test positive. Similarly, it seems that ABA does it in a negative way (see ref. 6). For this reason, we have never maintained that phytomelatonin can, exclusively, be the molecule responsible for these effects in physiological conditions. In this sense, the role of phytomelatonin on foliar senescence has been demonstrated. Melatonin down-regulates several chlorophyll degrading genes through the senescence factor SAG12, among others. The melatonin action occurs possibly cytokinin-mediated because melatonin up-regulates cytokinin level and some signaling elements, inhibiting the senescent process, in co-active melatonin-cytokinin collaboration (ref. 24). In any of the cases, we find a natural extract that, being rich in phytomelatonin, can behave with excellent regulatory properties of plant development, which are interesting to be applied in crops. However, the complete hormonal characterization of PTMA3 is one of our immediate goals. These considerations have been added to the text (Lines 345-356).

Round 2

Reviewer 1 Report

Required information has been added.